



# Quantifying methane point sources from fine-scale (GHGSat) satellite observations of atmospheric methane plumes

Daniel J. Varon[1,2], Daniel J. Jacob[1], Jason McKeever[2], Dylan Jervis[2], Berke O. A. Durak[2], Yan Xia[3], Yi Huang[3]

[1]School of Engineering and Applied Sciences, Harvard University, Cambridge, MA 02138, USA
[2]GHGSat, Inc., Montréal, QC H2W 1Y5, Canada
[3]Department of Atmospheric and Oceanic Sciences, Montréal, QC H3A 0B9, Canada

*Correspondence to*: Daniel J. Varon (danielvaron@g.harvard.edu)

**Abstract.** Anthropogenic methane emissions originate from a large number of relatively small point sources. The planned GHGSat satellite fleet aims to quantify emissions from individual point sources by measuring methane column plumes over selected $\sim$10×10 km$^2$ domains with $\leq$ 50×50 m$^2$ pixel resolution and 1-5% measurement precision. Here we develop algorithms for retrieving point source rates from such measurements. We simulate a large ensemble of instantaneous methane column plumes at 50×50 m$^2$ pixel resolution for a range of atmospheric conditions using the Weather Research and Forecasting model (WRF) in large eddy simulation (LES) mode and adding instrument noise. We show that standard methods to infer source rates by Gaussian plume inversion or source pixel mass balance are prone to large errors because the turbulence cannot be properly parameterized on the small scale of instantaneous methane plumes. The integrated mass enhancement (IME) method, which relates total plume mass to source rate, and the cross-sectional flux method, which infers source rate from fluxes across plume transects, are better adapted to the problem. We show that the IME method with local measurements of the 10-m wind speed can infer source rates with error of 0.07-0.17 t h$^{-1}$ + 5-12% depending on instrument precision (1-5%). The cross-sectional flux method has slightly larger errors (0.07-0.26 t h$^{-1}$ + 8-12%) but a simpler physical basis. For comparison, point sources larger than 0.5 t h$^{-1}$ contribute more than 75% of methane emissions reported to the U.S. Greenhouse Gas Reporting Program. Additional error applies if local wind speed measurements are not available, and may dominate the overall error at low wind speeds. Low winds are beneficial for source detection but not for source quantification.

## 1 Introduction

Conventional methane observing satellites are limited in their ability to detect individual point sources. They observe vertical columns of atmospheric methane by solar backscatter in the shortwave infrared (SWIR) with pixel resolutions of 1-10 km and column precisions of 0.1-1% (Bovensmann et al., 1999; Butz et al., 2011; Veefkind et al., 2012; Polonsky et al., 2014; Kuze et al., 2016). Jacob et al. (2016) show that these measurements are adequate for mapping regional methane emissions but cannot resolve individual methane point sources, which tend to be relatively small and spatially clustered (e.g., oil/gas fields, livestock operations, landfills, coal mine vents). The GHGSat microsatellite fleet (Germain et al., 2017; McKeever et al., 2017)





aims to address this gap by observing methane columns over selected scenes of order $10\times10$ km$^2$ with $\leq 50\times50$ m$^2$ effective pixel resolution and moderate precision (1-5%). Here we present an algorithm for interpreting the instantaneous plumes observed by such an instrument in terms of the implied point source (facility-level) emissions.

Aircraft remote sensing of methane columns over oil/gas and coal mining facilities reveals plumes with irregular shapes and detectable sizes of order 0.1-1 km (Thorpe et al., 2016; Thompson et al., 2015; 2016; Frankenberg et al., 2016). A standard method to retrieve source rates from plume observations is to assume Gaussian plume behavior, as expected from statistically averaged turbulence (Bovensmann et al., 2010; Krings et al., 2011; 2013; Rayner et al., 2014; Fioletov et al., 2015; Nassar et al., 2017; Schwandner et al., 2017). This method may induce large errors for small instantaneous plumes, which generally do not follow the steady-state Gaussian behavior. Krings et al. (2011, 2013) proposed a cross-sectional flux method

to derive the source rate as the product of the local wind and the concentration integrated over a plume cross-section, expanding on a similar method used for in situ plume measurements (White et al., 1976; Cambaliza et al., 2014; Conley et al., 2016). Jacob et al. (2016) described a mass balance method for inferring the source rate solely based on the enhancement in the source pixel. Frankenberg et al. (2016) inferred the source rate empirically from the total detectable mass of methane in the plume (integrated mass enhancement or IME).

A common feature of all these methods for retrieving the point source rate $Q$ from plume observations is their need for independent knowledge of the wind speed $U$ driving transport of the plume. In the cross-sectional flux method applied to in situ aircraft observations, methane and local wind speed are measured concurrently (Conley et al., 2016). In remote sensing, however, the wind speed for the instantaneous column plume is not directly measured and may be variable both vertically and horizontally across the plume.

Here we use observing system simulation experiments to develop an algorithm for retrieving individual point source rates from fine-scale satellite observations of methane plumes. We review previously-used plume inversion methods and show with large eddy simulations (LES) that the IME and cross-sectional flux methods are best-suited to the problem. We further develop the IME method to provide a physical basis for its general application. We consider different combinations of instrument error, meteorological environment, and wind information to test the methods and quantify errors. Our work is

motivated by the need to interpret GHGSat observations but is more generally applicable to any fine-scale plume observations from space.

## 2 Review of methods for retrieving point sources from observations of column plumes

A point source of methane produces a turbulent plume with characteristics determined by the strength of the source, the wind field, and turbulence that depends on atmospheric stability and surface roughness. Four different methods have been proposed

to quantify point source rates from plume observations: (1) the Gaussian plume inversion method (Bovensmann et al., 2010; Krings et al., 2011; 2013; Rayner et al., 2014; Fioletov et al., 2015; Nassar et al., 2017; Schwandner et al., 2017), (2) the source pixel method (Jacob et al., 2016; Buchwitz et al., 2017), (3) the cross-sectional flux method (White et al., 1976; Conley et al., 2016; Krings et al., 2011; 2013; Tratt et al., 2011; 2014; Frankenberg et al., 2016), and (4) the IME method (Thompson et al.,



2016; Frankenberg et al., 2016). Here we discuss these methods for remote sensing observations of column plumes, but similar methods are also used for interpreting in situ observations. In situ observations benefit from a stronger signal but require characterization of the plume in the vertical dimension, which is integrated in a column measurement.

Satellite remote sensing of methane plumes retrieves column concentrations with vertical sensitivity that depends on
atmospheric scattering and absorption. Clear-sky observations in the SWIR have near-unit sensitivity throughout the tropospheric column while observations in the thermal infrared (TIR) have strong vertical dependence determined by temperature contrast with the surface (Worden et al., 2013). Here we focus on SWIR observations, where we can ignore vertical dependence in sensitivity. TIR remote sensing has been used effectively to detect methane plumes from low-flying aircraft (Tratt et al., 2014; Frankenberg et al., 2016) but is not practical from space because of interference from the background
methane column above the plume (Jacob et al., 2016).

Methane column concentrations retrieved from remote sensing are commonly expressed in the literature as column average dry molar mixing ratio $X$ [ppb]. The plume is then characterized by an enhancement $\Delta X = X - X_b$ relative to the local background $X_b$. For our purposes of relating plume observations to the source rate $Q$ [kg s$^{-1}$], a more useful measure of plume concentration is the column mass enhancement $\Delta\Omega$ with units [kg m$^{-2}$]. $\Delta\Omega$ is related to $\Delta X$ by

$$\Delta\Omega = \frac{M_{CH_4}}{M_a} \Omega_a \Delta X , \tag{1}$$

where $M_{CH_4}$ and $M_a$ are the molar masses of methane and dry air [kg mol$^{-1}$] and $\mathbf{\Omega_a}$ is the column of dry air [kg m$^{-2}$].

**2.1 Gaussian plume inversion method**

The Gaussian plume inversion method fits a Gaussian plume model to the measured columns. Assuming a steady wind $U$ oriented along the $x$-axis and integrating the three-dimensional Gaussian plume equation vertically, one obtains an expression
for $Q$ in terms of the vertical column enhancement $\Delta\Omega(x,y)$ downwind of a point source located at the origin (Bovensmann et al., 2010):

$$Q = U\Delta\Omega(x,y)\left(\sqrt{2\pi}\sigma_y(x) \, e^{\frac{y^2}{2\sigma_y(x)^2}}\right). \tag{2}$$

The empirical dispersion parameter $\sigma_y(x)$ [m] describes the horizontal spread of the plume along the $y$-axis orthogonal to the wind direction. It is commonly parameterized as (Martin, 1976)

$$\sigma_y(x) = a\left(\frac{x}{x_0}\right)^{0.894} , \tag{3}$$

where $x_0 = 1000$ m and the dispersion coefficient $a$ [m] depends on atmospheric stability as defined by the Pasquill-Gifford atmospheric stability categories (Pasquill, 1961). The solution to (2) may involve non-linear optimal estimation fitting of $a$ to



the observed plume (Krings et al., 2011). The fit may not be successful if the instantaneous plume shows large departure from the steady-state Gaussian behavior, as is apparent for fine-scale methane plumes (Frankenberg et al., 2016).

**2.2 Cross-sectional flux method**

In the source pixel method used by Jacob et al. (2016) to compare different satellite observing configurations, emissions are

inferred solely from methane enhancements in the source pixel relative to the local background. For an observation pixel of dimension $W$ [m] containing a methane point source ventilated by a uniform wind speed $U$ [m s$^{-1}$], the source rate $Q$ [kg s$^{-1}$] is calculated from the mean source pixel enhancement $\Delta\Omega$ [kg m$^{-2}$]:

$$Q = \frac{UWp}{g\Omega_a}\Delta\Omega \; , \tag{4}$$

where $p$ is the surface pressure and $g$ is the acceleration of gravity. The source pixel method ignores additional information

from the plume downwind and is therefore not optimal. In addition, the instantaneous wind $U$ may have large uncertainty for small pixels because of turbulence. The method is also vulnerable to systematic errors in the column enhancement retrieved over the source pixel (e.g., due to different reflectance properties of the emitter compared to the surrounding area) and errors in the local background estimate.

**2.3 Cross-sectional flux method**

In the cross-sectional flux method, the source rate is estimated by computing the flux through one or more plume cross-sections orthogonal to the plume axis. This approach is commonly used for aircraft in situ observations (White et al., 1976; Mays et al., 2009; Cambaliza et al., 2014; 2015; Conley et al., 2016). Krings et al. (2011, 2013) and Tratt et al. (2011, 2014) extended it to methane columns observed by aircraft remote sensing. By mass balance, the source rate $Q$ must be equal to the product of the wind speed and the column plume transect along the $y$-axis perpendicular to the wind:

$$Q = \int_{-\infty}^{+\infty} U(x,y)\Delta\Omega(x,y)dy \; , \tag{5}$$

where the integral is approximated in the observations as a discrete summation of the product $U(x,y)\Delta\Omega(x,y)$ over the detectable width of the plume.

Compared to in situ aircraft measurements, an advantage of remote sensing is that the full vertical extent of the plume is covered by the measurement. A disadvantage is that the wind $U(x,y)$ is not as well characterized: it must describe some

vertical average over the plume extent and there is generally no information on its horizontal variability over the scale of the plume. This may require estimation of an effective wind speed $U_{eff}$ applied to the cross-plume integral $C$ [kg m$^{-1}$] of the column along the $y$-axis:

$$Q = CU_{eff} \; , \text{ with } C = \int_{-\infty}^{+\infty} \Delta\Omega(x,y)dy \; . \tag{6}$$





If $U_{eff}$ is assumed uniform with distance $x$ downwind of the source, then the integral $C$ is independent of $x$ and can be computed for different values of $x$ to improve accuracy through averaging. We show in Sect. 6 how to estimate $U_{eff}$ for use in Eq. (6).

**2.4 Integrated mass enhancement (IME) method**

The IME method relates the source rate to the total plume mass detected downwind of the source. The IME of an observed column plume comprising $N$ pixels of area $A_j$ ($j = 1 \dots N$) is

$$\text{IME} = \sum_{j=1}^{N} \Delta\Omega_j A_j \; . \tag{7}$$

Frankenberg et al. (2016) defined an empirical linear relationship between IME and $Q$ for their methane plumes detected from aircraft by using independent estimates of a few sources from the cross-sectional flux method. They then applied this linear
relationship to all their observed plumes.

More fundamentally, the relationship between IME and $Q$ is defined by the residence time $\tau$ of methane in the detectable plume: $Q = \text{IME}/\tau$. One can express $\tau$ dimensionally in terms of an effective wind speed $U_{eff}$ [m s$^{-1}$] and a plume size $L$ [m]:

$$Q = \frac{1}{\tau}\text{IME} = \frac{U_{eff}}{L}\text{IME} = \frac{U_{eff}}{L}\sum_{j=1}^{N} \Delta\Omega_j A_j \; . \tag{8}$$

$U_{eff}$ and $L$ would have simple physical meanings of wind speed and plume length if dissipation of the plume occurred by transport to a terminal distance downwind of the source. But the actual mechanism for plume dissipation is turbulent diffusion, which takes place in all directions. $U_{eff}$ and $L$ must therefore be viewed as operational parameters to be related to observations of wind speed $U$ and plume extent. In Sect. 5 we derive these relationships from synthetic plumes generated by large eddy simulation (LES). The detectable plume size $L$ depends on $Q$ and $U$, introducing non-linearity in Eq. (8).

**3 Synthetic GHGSat observations of methane plumes**

We generate synthetic GHGSat plumes with the Weather Research and Forecasting model in LES mode (WRF-LES; Moeng et al., 2007) to evaluate the ability of the methods described in Sect. 2 to retrieve methane point source rates from satellite observations. The WRF-LES simulations are conducted at 50×50 m$^2$ resolution and are sampled virtually with the GHGSat instrument by column integration and with consideration of instrument precision. In this section, we briefly describe the
GHGSat instrument and the application of LES to produce synthetic plumes.

**3.1 The GHGSat instrument**

GHGSat is a lightweight satellite instrument (~15 kg) designed by GHGSat, Inc. to detect atmospheric methane plumes from individual point sources. A demonstration instrument (GHGSat-D) was launched in June 2016 into sun-synchronous orbit





(local solar viewing time of 09:30 on the descending node) to test the instrument performance and column retrieval algorithm. The launch of the first operational satellite is scheduled for early 2019. The long-term plan is for a constellation of sun-synchronous GHGSat microsatellites in low Earth orbit providing frequent observations of different sources of interest. GHGSat measures backscattered solar SWIR radiation at 1635-1670 nm (0.1 nm spectral resolution) over ~10×10 km$^2$

targeted domains with ≤ 50×50 m$^2$ pixels. The target precision for the methane column retrieval is 1-5%.

### 3.2 Large eddy simulations (LES)

We apply WRF-LES to simulate turbulent plume transport at 50×50 m$^2$ horizontal resolution and 30-m vertical resolution over a 6×6 km$^2$ domain. We use a modified version of the WRF v3.8 default LES case with cloud-free conditions and no topography (WRF User Guide; Moeng et al., 2007). Simulations are performed with one-way nesting from an external

simulation over a 7.5×7.5 km$^2$ domain with 150×150 m$^2$ resolution and periodic boundary conditions. A uniform sensible heat flux $H = 100$ W m$^{-2}$ is applied at the surface to drive buoyant turbulence and mechanical turbulence is generated by surface drag with an aerodynamic roughness height of 0.1 m. Forcing from a large-scale pressure gradient maintains momentum across the domain.

Each LES simulates five hours of atmospheric transport. The first three hours spin up realistic turbulence and the final

two hours are used for analysis. We use a range of initial mixing depths and wind speed soundings to produce different simulations. The potential temperature soundings are uniform at 290 K from the surface to a mixing depth set at either 500, 800, or 1100 m altitude, with an inversion above that altitude and the model top set 700 m above the inversion. For each of these three mixing depths, we conduct simulations using five initially uniform southerly wind profiles with speeds of 2-8 m s$^{-1}$. The resulting LES ensemble of 15 simulations is broadly representative of the range of meteorological conditions that could

be sampled with a SWIR daytime instrument.

We use the WRF-LES passive tracer transport capability (Nottrott et al., 2014; Nunalee et al., 2014) to generate a plume from a single constant point source in the WRF-LES meteorological environment. From there we integrate the plume over vertical columns and add random noise to produce GHGSat pseudo-observations. We archive the tracer column field every 30 seconds as an independent realization of the instantaneous plume. From the 15 WRF-LES simulations we thus archive

a collection of 3600 scenes, representing our statistical ensemble for different possible realizations of turbulence.

The WRF-LES point source in the archived ensemble has a normalized source rate, which we subsequently scale from the output to simulate source rates $Q$ in the range 0.05-2.25 t CH$_4$ h$^{-1}$ (0.5-20 kt a$^{-1}$). This range covers the top 25% of sources reporting to the U.S. Greenhouse Gas Reporting Program (GHGRP) and contributing 80% of total GHGRP methane emissions (Jacob et al., 2016). A uniform background methane column of 0.01 kg m$^{-2}$ (roughly 1850 ppb) is added to the tracer

column. Uncorrelated measurement noise is then added as a random increment of the background sampled from a normal distribution with zero mean bias and standard deviation $\sigma = 1$-5%, corresponding to the range of expected instrument precision. The column enhancement $\Delta\Omega$ is then inferred by subtracting the 0.01 kg m$^{-2}$ background, which is therefore assumed to be known.



Figure (1) shows examples of synthetic plume observations produced in this manner for a source $Q = 1$ t h$^{-1}$, assuming different levels of instrument precision. As the noise standard deviation increases, the simulated plume becomes increasingly difficult to detect.

**4 Inadequacy of the Gaussian plume and source pixel methods**

Previous studies of $CO_2$ column observations from the OCO-2 satellite instrument with $\sim 1.3 \times 2.25$ km$^2$ nadir pixel resolution (Crisp et al., 2017) have shown that the Gaussian plume inversion method can be effective for quantifying very large $CO_2$ emissions from power plants (Nassar et al., 2017) and volcanoes (Schwandner et al., 2017). However, we find that the approach fails when applied to fine-scale methane plumes because they depart too much from Gaussian behavior. $CO_2$ point sources can be considerably larger relative to background concentrations and instrument precision levels, and the

resulting plumes can then be observed over distances of tens of kilometers. Such large size allows for statistical averaging of eddies and hence better Gaussian behavior even for an instantaneous plume. To demonstrate this, Figure (2) shows an LES snapshot of a large power plant emitting 3.75 kt $CO_2$ h$^{-1}$ in a 72-km domain with $300 \times 300$ m$^2$ pixel resolution. Fitting a Gaussian plume to the 300-m pixel enhancements yields a coefficient of determination R$^2$ of only 0.24, but R$^2$ increases to 0.86 when the LES image is averaged over $3 \times 3$ km$^2$ pixels. Spatial averaging of turbulence over kilometer-scale pixels thus

greatly improves the accuracy of Gaussian plume models, but this requires sufficiently large plumes. Methane plumes requiring 1-5% precision for detection are too small to allow such averaging (Frankenberg et al., 2016).

    The source pixel retrieval method only considers the column enhancement over the point source pixel, thus inferring the source rate from ventilation of that pixel by the local wind. It assumes in effect that the near-field plume is diluted over the source pixel and neglects information from the plume downwind. This can be an effective method when pixel resolution is

coarse, so that most of the information is in the source pixel and the mean wind across the pixel can be well-defined (Buchwitz et al., 2017). However, it has three major shortcomings when applied to GHGSat $50 \times 50$ m$^2$ pixels: (1) it does not exploit the information from downwind pixels, where most of the plume mass typically resides; (2) small-scale turbulence generates strong variability in the wind; and (3) source pixel ventilation may take place by turbulent horizontal diffusion rather than advection by the mean wind, leading to negative bias. With regard to (2), the residence time in a GHGSat pixel is only $\sim 30$ s,

and there is large variability in the wind on such a short time scale that cannot be described deterministically. For example, in a typical LES under moderately unstable conditions we find a 10-m wind speed of $2.45 \pm 0.8$ m s$^{-1}$, where the standard deviation is for the 30-s data. 30-second variability in wind speed alone thus introduces a factor of 30% uncertainty in the source estimate. With regard to (3), the relative importance of turbulent diffusion and advection is diagnosed by the Péclet number Pe $= UL/K_H$, where $K_H$ is the turbulent horizontal diffusion coefficient (Brasseur and Jacob, 2017). For a typical

$K_H = 100$ m$^2$ s$^{-1}$ (d'Isidoro et al., 2010) with $U = 2$ m s$^{-1}$ and $L = 50$ m we find Pe $\sim 1$, so that turbulent diffusion and advection are of comparable importance.





## 5 Computing the source rate by the IME method

We showed in Sect. 2.4 how the IME method for retrieving the point source rate $Q$ from the measured IME hinges on knowledge of the residence time of methane in the detectable plume. We refer to this residence time as the plume lifetime $\tau = \text{IME}/Q$, which in turn is related to two parameters: an effective wind speed $U_{eff}$ and a characteristic plume size $L$. IME

and $L$ can be inferred from the plume observations, while $U_{eff}$ can be inferred from the observable 10-m wind speed $U_{10}$ at the point of emission.

### 5.1 Inferring the plume mass (IME) and size ($L$)

Inferring IME and $L$ from the plume observations requires that we define the horizontal extent of the plume through a pixel selection procedure that separates signal from noise. Careful selection is important. Consider an array of $N$ pixels of equal area

and with retrieved column enhancements $\Delta\Omega_j$ ($j = 1, …, N$). If each pixel enhancement includes a contribution $s_j$ from signal (actual plume enhancement) and $\varepsilon_j$ from random noise, then as per Eq. (7),

$$\frac{\text{IME}}{A} = \sum_{j=1}^{N} \Delta\Omega_j = \sum_{j=1}^{N}(s_j + \varepsilon_j) = \varepsilon_a + \sum_{j=1}^{N} s_j \,, \tag{9}$$

where $\varepsilon_a$ is the total measurement error. The relative error $\varepsilon_r$ is then $\varepsilon_r = \varepsilon_a / \sum_j^N s_j$. If the noise is normally distributed and uncorrelated, then the error standard deviation is proportional to $\sqrt{N}$, so that the standard deviation $\sigma_r$ of the relative error

scales as

$$\sigma_r \propto \frac{\sqrt{N}}{\sum_j^N s_j} \,. \tag{10}$$

Now consider two extreme cases: (1) all pixels contain the same signal $s_0$, and (2) only one pixel contains signal $s_0$ and the other pixels contain only noise. In case (1), the total signal $\sum_j^N s_j$ is proportional to $N$, meaning $\sigma_{\varepsilon_r} \propto 1/\sqrt{N}$. By contrast, in case (2), the total signal is equal to $s_0$, so $\sigma_{\varepsilon_r} \propto \sqrt{N}$. Thus, we see that aggregating plume pixels can either decrease or increase

the error on the IME depending on whether these pixels have significant signal or not.

Figure (3) illustrates how we construct a plume mask to select plume pixels with significant signal-noise ratios. The background distribution (mean ± standard deviation) is first characterized by an upwind sample of the measured columns, mimicking what one would do with actual observations. Next, we sample the 5×5 pixels neighborhood centered on each pixel in the viewing domain and compare the sample distributions to the background distribution by means of a Student's $t$-test.

Pixels whose 5×5 neighborhoods follow a distribution significantly different than the background at a significance level of 95% or higher are assigned to the plume, others to the background. The resulting Boolean plume mask contains some random classification errors, so we smooth it with a 3×3 (binary-valued) median filter followed by a Gaussian filter and thresholding.

We compute the IME by summing pixel enhancements within the plume mask following Eq. (7). A simple measure of the plume size $L$ can be made as





$$L = \sqrt{A_M} \, , \tag{11}$$

where $A_M$ [m$^2$] is the area of the plume mask. Another possible estimate of $L$ would be the mask's perimeter, which can be obtained by contour-tracing. The definition of $L$ is not critical as long as it has some physical basis relating it to the observed plume geometry. A different definition would imply a different calculation of $U_{eff}$.

## 5.2 Inferring $U_{eff}$ from the 10-m wind speed $U_{10}$

The effective wind speed $U_{eff}$ is a parameter of the IME method that should be related to the measurable 10-m wind speed at the location of the point source, and here we use the LES to derive the $U_{eff} = f(U_{10})$ relationship. If $U_{10}$ is not actually measured at the site, it can be estimated from an operational meteorological database at the cost of some representation error. We discuss that error in Sect. 7.

10       We derive the $U_{eff} = f(U_{10})$ relationship from a training set of column plumes comprising two thirds of the LES ensemble selected at random (i.e., 2400 plume instances). The remaining plumes serve as a test set for evaluating the retrieval algorithm. For each plume in the training set, $U_{eff}$ is computed from Eq. (8) as $U_{eff} = QL/\text{IME}$, based on the known source rate $Q$ and with $L$ and IME determined from the plume masks. The corresponding $U_{10}$ time series at the location of the source is obtained from the LES, averaged over the plume lifetime $\tau = \text{IME}/Q$. Values of $\tau$ in our ensemble range from 1 to 60

minutes depending on instrument precision, source rate, and wind speed. In practice, $\tau$ is unknown *a priori* and must be inferred from the plume observations and local wind speed information. We discuss this in Sect. 5.3.

      Figure 4 shows the relationship between $U_{eff}$ and $U_{10}$ inferred from the LES ensemble. We find that we can fit the data to a logarithmic dependence:

$$U_{eff} = \alpha_1 \log U_{10} + \alpha_2 \, , \tag{12}$$

where $\alpha_1 = 1 \pm 0.1$ m s$^{-1}$, $\alpha_2 = 0.55 \pm 0.05$ m s$^{-1}$, and the ranges on the coefficients are for the 1-5% range of instrument precision. For 1% instrument precision, the logarithmic function plotted in Figure 4 captures 75% of the variance ($R^2 = 0.75$). This decreases to 35% of the variance for 5% instrument precision. The convexity of the relationship is an important result, as it implies that error in $U_{eff}$ is smaller than error in $U_{10}$. One might expect $U_{eff}$ from the IME method to be proportional to $U_{10}$, such that IME/$L$ would be inversely proportional to $U_{10}$ as per Eq. (8). However, even though that inverse relationship

holds for plume concentrations (see Eq. (6)), it is much weaker for the IME because the concentrated plume in the near-field of the source remains in the signal even at high wind speeds. Thus, the plume observations themselves interpreted with the IME method contain some information on $U_{10}$ that slackens the dependence of $U_{eff}$ on $U_{10}$.



### 5.3 Computing the source rate

Figure (5) summarizes the algorithm for retrieving source rates with the IME method. The algorithm accepts two inputs: (1) a map of plume enhancements $\Delta\Omega(x, y)$ over the plume mask, and (2) the 10-m wind speed $U_{10}$ from either local high-frequency measurements or an operational meteorological database. If local high-frequency measurements of $U_{10}$ are available, then

there is an opportunity to iteratively refine the plume lifetime $\tau$ over which $U_{10}$ should be averaged and for this we make a first guess $\tau_0 = 5$ minutes. If only coarse-resolution wind speed data are available, then we assume that these are representative of the local value averaged over the plume lifetime and add the associated error to the overall error budget (see Sect. 7).

Figure (6) shows the results of our IME retrieval algorithm when applied to the test set of LES plumes in different instrument precision scenarios. In all cases, the source rate predictions show good agreement with the 1:1 line ($R^2 \geq 0.86$).

Retrieval uncertainty (expressed as absolute + relative contributions and defined by the standard deviation of departure from the 1:1 line) increases from 0.07 t h$^{-1}$ + 5% for $\sigma = 1\%$, to 0.13 t h$^{-1}$ + 7% for $\sigma = 3\%$, and 0.17 t h$^{-1}$ + 12% for $\sigma = 5\%$ (Table 1). For sources 1.5 t h$^{-1}$ or larger, retrieval error is less than 25% even with instrument precision up to 5%. For $Q = 1$ t h$^{-1}$, instrument precision up to $\sigma = 3\%$ yields uncertainty less than 20% of the true source rate. Source rates larger than 0.5 t h$^{-1}$ contribute ~75% of total methane emitted from point sources reporting to the U.S. Greenhouse Gas Reporting Program

(Jacob et al., 2016). An instrument with $\sigma = 1\%$ measurement uncertainty can quantify these emissions to within 20% of the true source rate.

### 6 Computing the source rate by the cross-sectional flux method

Much of our analysis of the IME method in Sect. 5 can be applied to the cross-sectional flux method commonly used for in situ aircraft observations, and extended by Krings et al. (2011, 2013) and Tratt et al. (2011, 2014) for remote sensing

observations. We compute the plume mask as described in Sect. 5.1, and infer the wind direction from the axis of the plume, based on a weighted average of plume pixel coordinates using the column enhancements as weights. From there, we obtain the mean cross-plume integral $C$ of the column enhancements at different distances downwind of the source (see Eq. (6)).

We again use the LES training set to characterize the relationship between the effective wind speed $U_{eff}$ in Eq. (6) and the local 10-m wind speed $U_{10}$. For each plume in the training set, $U_{eff}$ is computed from Eq. (6) based on $C$ and the

known source rate $Q$. The plume lifetime over which to average local high-frequency $U_{10}$ measurements for comparison with $U_{eff}$ is computed as $\tau = L/U_{eff}$, where the plume size parameter $L$ now has a specific physical meaning as the maximum along-wind distance from the source over which transects can be computed (as defined by the plume mask).

Figure (7) shows the resulting relationship between $U_{eff}$ and $U_{10}$. The relationship is near-linear, as would be expected, and the fit $U_{eff} = \beta\,U_{10}$ with $\beta = 1.44 \pm 0.04$ (where the range is for the 1-5% range of instrument precisions)

captures 20-77% of the variance ($0.20 \leq R^2 \leq 0.77$) for $U_{10} \geq 2$ m s$^{-1}$, depending on instrument precision. The ~44% increase relative to $U_{10}$ reflects the increase of wind speed with altitude where the plume is transported. The departure from





the linear relationship for $U_{10} < 2$ m s$^{-1}$ is because low winds are more variable in direction. The cross-sectional flux method should not be used under calm wind conditions.

Figure (8) shows the results of the cross-sectional flux retrieval algorithm applied to the LES test plumes, excluding those from the plume population with $U_{10} < 2$ m s$^{-1}$ and $U_{eff} < 2$ m s$^{-1}$. In all instrument precision scenarios, the retrieved

source rates are consistent with the 1:1 line. However, residuals are slightly larger than in the IME method (see Figure (6)), as indicated by the smaller coefficients of determination. This results primarily from greater uncertainty in the effective wind speed compared to the IME method. Moreover, analyzing orthogonal plume cross-sections requires estimation of the wind direction, which introduces an additional source of error. Absolute and relative retrieval errors estimated in the same way as for the IME method are listed in Table 1. While retrieval uncertainty is slightly higher (0.07-0.26 t h$^{-1}$ + 8-12%, depending on

instrument precision), an advantage of the cross-sectional flux method is that there is a simpler physical basis for relating $U_{10}$, $C$, and $Q$.

**7 Inferring the effective wind speed from meteorological databases**

Both the IME and cross-sectional flux methods require knowledge of the local wind speed. In the absence of local wind speed measurements, the 10-m wind speed $U_{10}$ at the time of observation must be estimated from some meteorological database.

Here we examine the option of using the GEOS-FP operational reanalysis produced by the NASA Global Modeling and Assimilation Office, available globally as 3-hour averages with 0.25°×0.3125° resolution ($\approx$ 25×25 km$^2$) at a lowest gridpoint level of 60 m above the surface (Molod et al., 2012; https://gmao.gsfc.nasa.gov/GMAO_products/). The 10-m wind speed can be obtained from the 60-m wind speed by:

$$U_{10} = \left[ \frac{\ln\left(\frac{z_{10}}{z_{0,m}}\right) - \Psi_m}{\ln\left(\frac{z_{60}}{z_{0,m}}\right) - \Psi_m} \right] U_{60} , \tag{13}$$

where $z_{0,m}$ [m] is the surface roughness length for momentum, $z_{10} = 10$ m, $z_{60} = 60$ m, and $\Psi_m = f(z/l)$ is a stability correction parameter dependent on the Monin-Obukhov length $l$ (Brasseur and Jacob, 2017). The GEOS-FP data include values for $z_{0,m}$ and $l$, but one can use local estimates of these variables if better information is available. Better databases than GEOS-FP may be available to the user depending on region, but an advantage of GEOS-FP is that it can be used as a global default.

Figure (9) evaluates the GEOS-FP $U_{10}$ data by comparison to 3-hour average daytime measurements in June 2017 at 10 U.S. airports obtained from the University of Utah MesoWest database (http://mesowest.utah.edu/). Here we use $z_{0,m} = 0.025$ m as input to Eq. (13) to account for the relatively smooth airport terrain. There is no bias in the GEOS-FP data relative to MesoWest. The error standard deviation derived from the difference between the 3-hour GEOS-FP and MesoWest 10-m wind speeds is 1.6 m s$^{-1}$, largely independent of wind speed. Since wind speed is a positive variable, errors at low wind speeds

(< 2 m s$^{-1}$) tend to be systematic. There is additional error from using 3-hour wind data when the plume lifetime $\tau$ is much



shorter. From the 5-minute resolution of the MesoWest data we find an additional error standard deviation of 2.0 m s$^{-1}$ for $\tau =$ 5 minutes and 1.3 m s$^{-1}$ for $\tau = 1$ hour when 3-h average wind speed data are used. Adding these errors in quadrature, we conclude that using GEOS-FP wind data incurs an error standard deviation on the 10-m wind speed of 2.5 m s$^{-1}$ for small plumes ($\tau = 5$ minutes) and 2.0 m s$^{-1}$ for large plumes ($\tau = 1$ hour).

5        Substitution into the $U_{eff} = f(U_{10})$ relations of the IME and cross-sectional flux methods implies an additional error in inferring $Q$ of 15-50% for the IME method and 30-65% for the cross-sectional flux method over the 10-m wind speed range 2-7 m s$^{-1}$, with largest errors at low wind speeds. The error is larger for the cross-sectional flux method where the dependence of $U_{eff}$ on $U_{10}$ is linear rather than logarithmic. Comparison to the other retrieval errors for each method is given in Table 1. At low wind speeds, the error associated with using GEOS-FP wind data may dominate the overall error budget for inferring

source rates. However, our estimate of the error from using operational meteorological databases is intended only to be illustrative. Different errors may apply for other regions or seasons, or when using other meteorological databases than GEOS-FP.

**8 Conclusions**

We have developed new algorithms for quantifying methane point sources from fine-scale satellite observations of atmospheric

column plumes, motivated by the planned fleet of GHGSat instruments ($\leq 50 \times 50$ m$^2$ pixel resolution, 1-5% precision). A challenge is that individual point sources of methane are relatively weak, so that the detectable instantaneous plumes are relatively small ($\sim$1 km) and short-lived ($<$ 1 hour). Using a large ensemble of WRF large eddy simulations (LES) of methane plumes from point sources, we showed that Gaussian plume inversions are unsuccessful because the instantaneous plumes are too small to follow Gaussian behavior. We also showed how a simple source pixel mass balance method is inappropriate

because of wind variability and horizontal turbulent diffusion on the scales of relevance.

Two more promising methods for quantifying source rates from methane column plume observations are the integrated mass enhancement (IME) method and the cross-sectional flux method. Both methods require a carefully constructed plume mask to isolate the plume enhancements from the background noise. The IME method requires estimation of the plume lifetime $\tau$, which in turn depends on an effective wind speed $U_{eff}$ for the plume and a characteristic plume size $L$. We showed

how these quantities can be estimated from knowledge of the plume mask and of the 10-m wind speed $U_{10}$ at the location of the source. The source rates are then inferred from the plume observations with expected errors of 0.07-0.17 t h$^{-1}$ + 5-12% depending on instrument precision (1-5%). For reference, source rates larger than 0.5 t h$^{-1}$ contribute 75% of total point source emissions in the U.S. Greenhouse Gas Reporting Program (GHGRP) database.

The cross-sectional flux method requires an estimate of the wind direction and of an effective wind speed $U_{eff}$

reflecting the vertical and horizontal spread of the plume. Again, the LES simulations show how these can be reliably estimated from the plume mask and local $U_{10}$. We find that for $U_{10} \geq 2$ m s$^{-1}$, $U_{eff} = \sim1.44 \, U_{10}$ is a good approximation to account for vertical plume spreading. The cross-sectional flux method should not be used for $U_{10} < 2$ m s$^{-1}$. The source rates are inferred





from the plume observations with expected errors of 0.07-0.26 t h$^{-1}$ + 8-12%, slightly worse than in the IME method and for a narrower range of 10-m wind speeds, but with a simpler physical basis.

Both the IME and the cross-sectional flux methods parameterize their effective wind speed $U_{eff}$ as a function of the local wind speed $U_{10}$. If local measurements of $U_{10}$ are not available, then $U_{10}$ must be estimated from an operational
meteorological database or from measurements some distance away. Using the global NASA GEOS-FP archive of wind speeds in June 2017 as an illustrative example compared to U.S. airport data, we find that using this archive would incur source rate errors of 15-50% in the IME method and 30-65% in the cross-sectional flux method over the 2-7 m s$^{-1}$ range of wind speeds. The largest errors are at low wind speeds where they dominate the overall error budget. Low wind speeds facilitate source detection by improving signal to noise, but worsen source quantification by increasing uncertainty in the inference of $U_{eff}$.

**Data availability**

The University of Utah MesoWest meteorological data used in this study is freely available at http://mesowest.utah.edu/. Similarly, the NASA GEOS-FP data is available from the GEOS-5 Data Server at https://portal.nccs.nasa.gov/cgi-lats4d/opendap.cgi?&path=. LES model output data is available upon request.

**Competing interests**

The authors declare that they have no conflict of interest.

**Acknowledgements**

DJJ's contribution was supported by the NASA Earth Science Division.

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





**Table 1. Error standard deviations for retrieving point source rates from column plume observations.**

| Method | Instrument precision[a] | | | Local wind speed estimate[b] |
|---|---|---|---|---|
| | 1% | 3% | 5% | |
| IME | $0.07$ t h$^{-1}$ + 5% | $0.13$ t h$^{-1}$ + 7% | $0.17$ t h$^{-1}$ + 12% | 15-50% |
| Cross-sectional flux | $0.07$ t h$^{-1}$ + 8% | $0.18$ t h$^{-1}$ + 8% | $0.26$ t h$^{-1}$ + 12% | 30-65% |

[a] Sum of absolute and relative errors when local measurements of 10-m wind speed $U_{10}$ are available (see text).

[b] Additional error when local wind speed data are not available, to be summed in quadrature with relative errors from the case where they
are available. The values given here are inferred from a sample of the GEOS-FP global database and should only be viewed as illustrative.
The range is for GEOS-FP wind speeds of 2-7 m s$^{-1}$, with the largest error values corresponding to the smallest wind speeds.




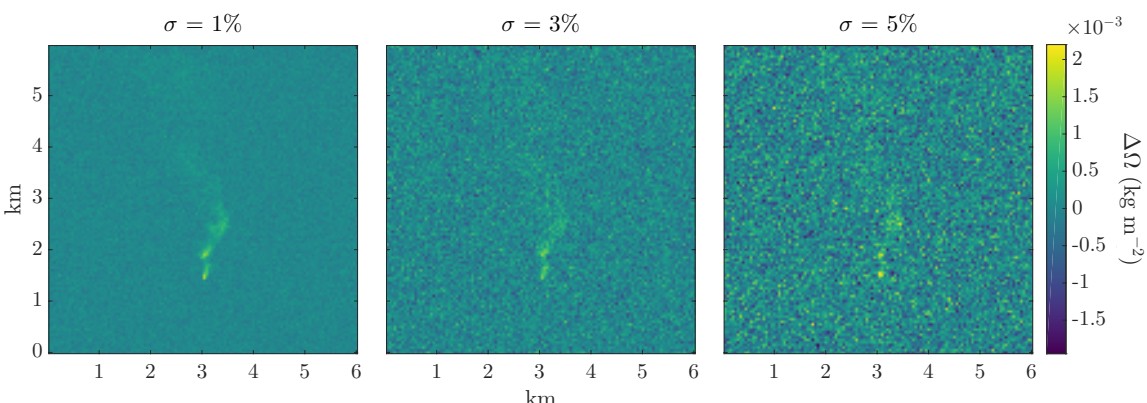

**Figure 1: Examples of column plume pseudo-observations generated by an LES on a 50×50 m$^2$ grid. Each panel shows the same synthetic plume observation for a source Q = 1 t h$^{-1}$, with instrument precision σ varying from 1% to 5%.**

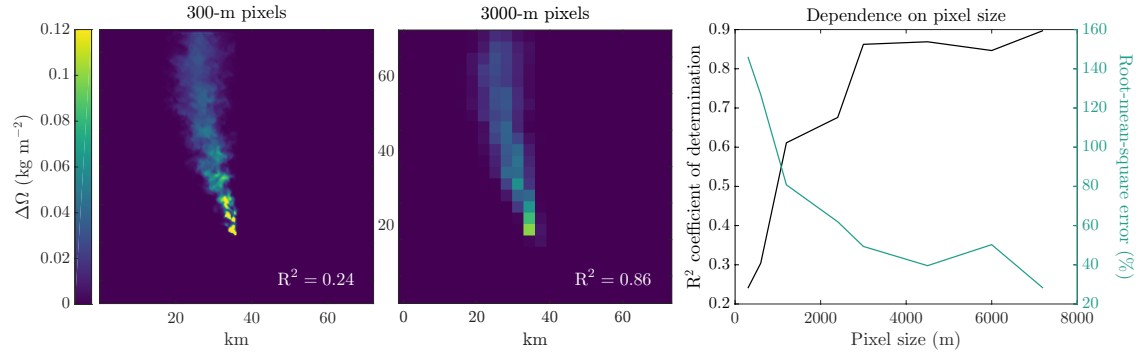

**Figure 2: CO$_2$ column enhancements relative to background for a 3.75 kt CO$_2$ h$^{-1}$ (33 Mt CO$_2$ yr$^{-1}$) power plant plume simulated by LES at 300-m resolution. The left panel shows the plume with 300-m pixel resolution and the middle panel shows the same plume but with pixel resolution degraded to 3000 m. The coefficient of determination (R$^2$) inset measures the ability to fit each LES plume to a Gaussian form (Eq. (2)-(3)). The right panel shows how the coefficient of determination and the root-mean-square error (expressed as a percentage of the median pixel enhancement) vary with pixel resolution.**



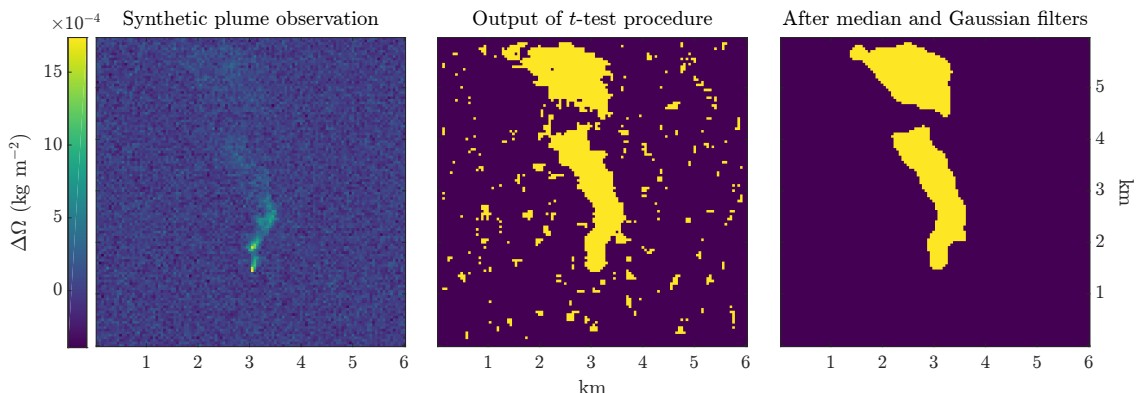

**Figure 3: Illustration of the procedure for constructing plume masks in the IME method. (Left) Satellite pseudo-observation generated by LES for a point source Q = 1 t h⁻¹, with instrument precision σ = 1% (same as in Figure 1). (Middle) The output of the t-test procedure for significant signal. (Right) The final plume mask after application of median and Gaussian filters.**





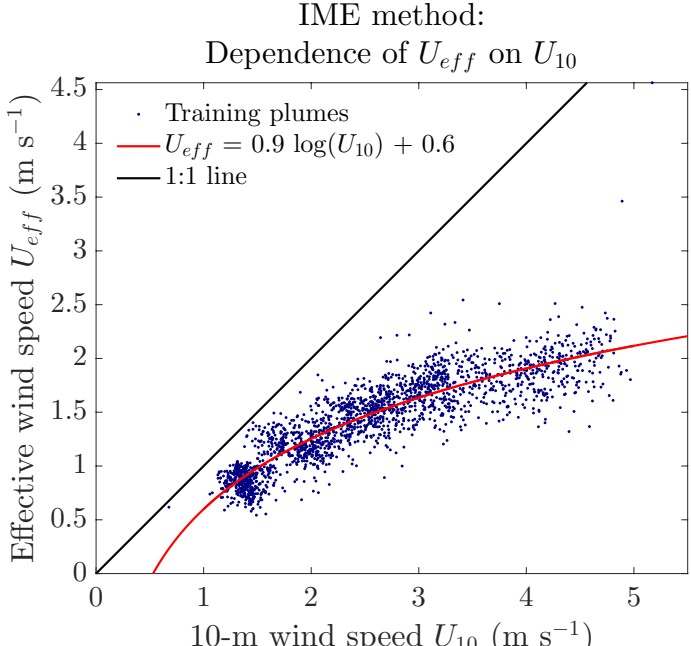

**Figure 4:** Relationship between the effective and local 10-m wind speeds in the IME method, characterized with LES training plumes assuming 1% instrument precision. Each point represents a different LES plume pseudo-observation from the training set. The red line fits the data to a logarithmic dependence. The 1:1 line is shown in black. See text for similar results with 3% or 5% instrument precision.



# Source rate retrieval by the IME method

Figure 5: Flow chart describing the IME retrieval algorithm. Algorithm inputs are shown in green, operations in grey, and output in blue. There are two possible paths depending on the availability of 10-m wind speed data: (a) local high-frequency wind speed measurements at the location of the source (right branch), and (b) a temporally averaged meteorological database (left branch).





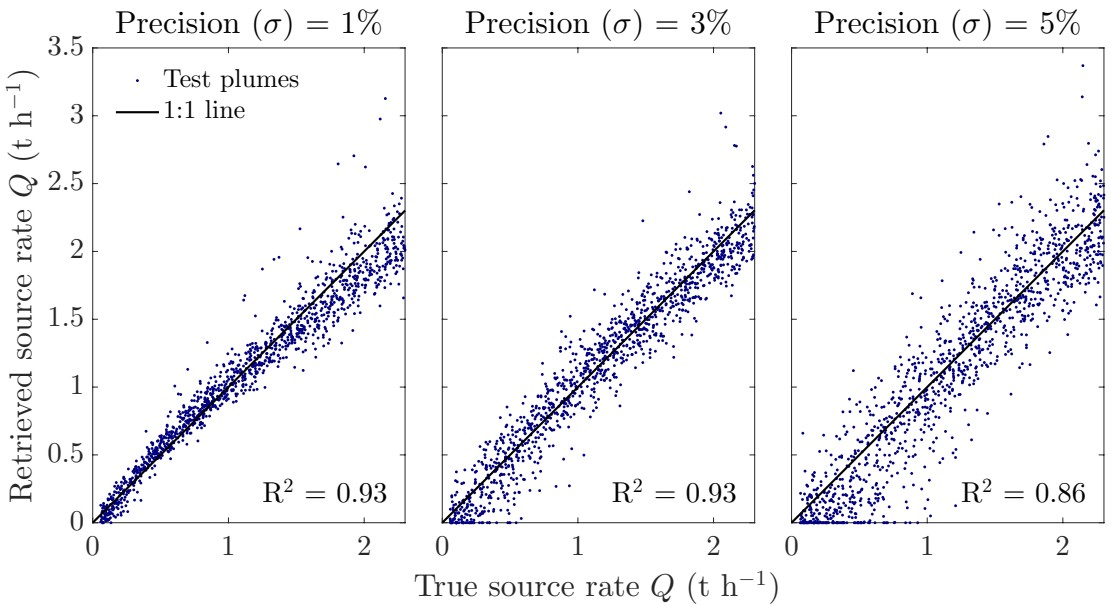

**Figure 6: Evaluation of the IME method for retrieving source rates Q using the LES test set with three different instrument precisions (1%, 3%, 5%). The inset gives the coefficient of determination, $R^2$, relative to the 1:1 line.**




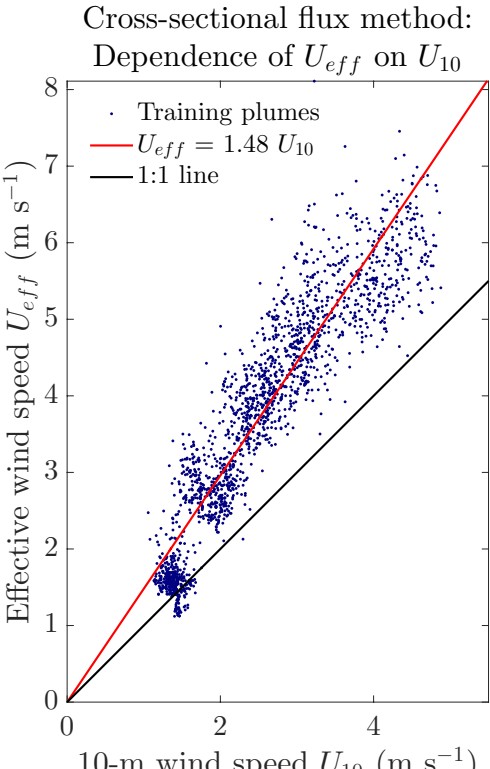

**Figure 7: Relationship between the effective and local 10-m wind speeds in the cross-sectional flux method, characterized with LES training plumes assuming 1% instrument precision. Each point represents a different LES plume pseudo-observation from the training set. The red line fits the data to a linear function, excluding the lowest wind speed population ($U_{10} < 2$ m s$^{-1}$ and $U_{eff} < 2$ m s$^{-1}$). See text for corresponding results with 3% and 5% instrument precision.**





## Testing the cross-sectional flux method

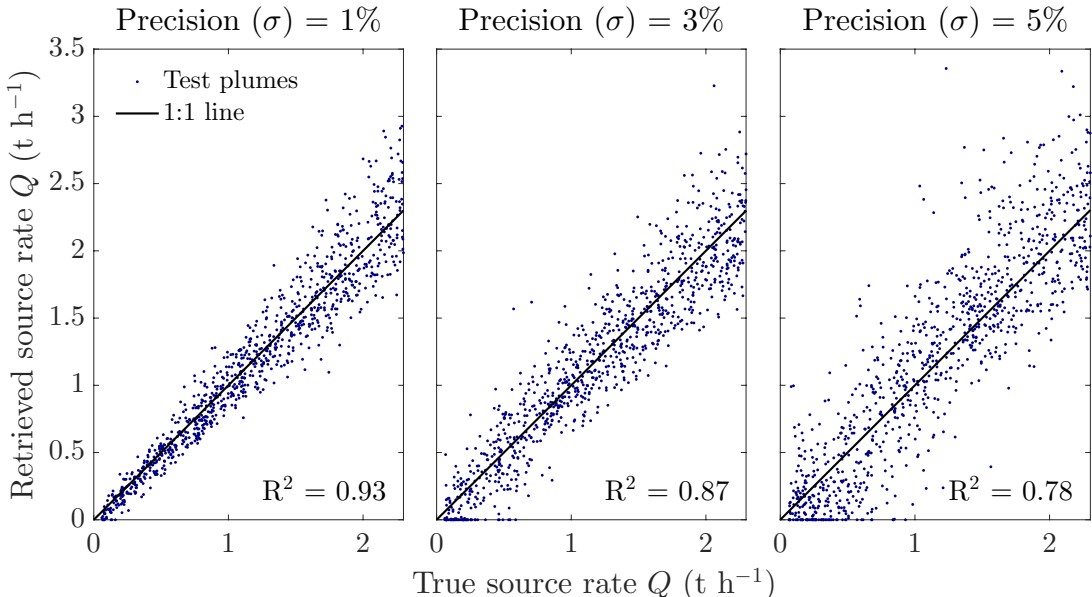

**Figure 8: Evaluation of the cross-sectional flux method for retrieving source rates Q using the LES test set with three different instrument precisions (1%, 3%, 5%). The inset gives the coefficient of determination, $R^2$, relative to the 1:1 line.**





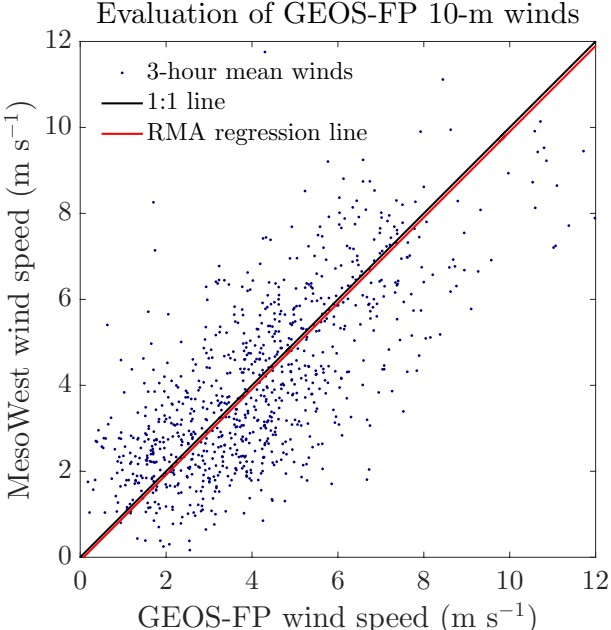

**Figure 9: Evaluation of 10-m wind speeds from the 3-hour GEOS-FP global database when used as estimate of local wind speed for source rate calculations in the IME and cross-sectional flux methods. The figure compares three-hour average 10-m wind speeds from the MesoWest database measured at 10 U.S. airports (ABQ, ATL, BOS, DFW, LAX, MCI, MSP, PDX, PHL, and PHX) to**
5  **corresponding values from the GEOS-FP database. The GEOS-FP data have been corrected for a local roughness height $z_{0,m} = 0.025$ m (see text). The data are for daytime June 2017 (15:00-21:00 UTC). The fit to a reduced major axis (RMA) regression line is also shown, which closely overlaps the 1:1 line.**