# Peer review of "Quantifying methane point sources from fine-scale satellite observations of atmospheric methane plumes"

_Atmospheric Measurement Techniques, 2018_

## Referee Comment (RC1) · Dr. Nassar (Referee) · 5 Jul 2018

"Quantifying methane point sources from fine-scale (GHGSat) satellite observations of atmospheric methane plumes" by Varon et al. compares four different approaches for methane point source quantification using simulated data. The two most promising methods (Integrated Mass Enhancement and Cross Sectional Flux) for this application are described in more detail along with characterization of their uncertainties. This short manuscript provides some very useful results that are timely and widely applicable to the current state of science in this field. While the launch of GHGSat-D clearly motivated this work, with no actual GHGSat data used in the study and multiple potential missions on the horizon in the same range of capabilities, to make this work most widely applicable to the scientific community, it would be advisable to keep the paper's title more general (removing 'GHGSat'). This would be consistent with the authors' statement on page 2 "Our work is motivated by the need to interpret GHGSat observations but is more generally applicable to any fine-scale plume observations from space".

Overall, my view is that the findings of the study are scientifically sound and generally justified by the simulations. The study also demonstrates some of the important differences in $CH_4$ and $CO_2$ point source quantification that we not so apparent even just a few years ago, but are becoming clearer with the heightened scientific attention to this field. I would recommend acceptance of the manuscript for publication in AMT with the suggested modification to the title and provided that a number of specific points outlined below could be addressed.

Specific Comments and Technical Corrections

Page 1, Line 25: With such a small number of nadir methane column observing satellites, which use a diversity of technologies that result in a range of observing characteristics, the word "conventional" does not really apply. "Most existing and upcoming methane observing satellites ..." would be a better introductory phrase.

P1, L26: Since Jacob et al. (2016) reviews methane observations from space, the authors could easily have provided a more accurate description of SWIR mission pixel resolutions here than "1-10 km". From the list in Jacob et al. (2016) the proposed CarbonSat has the smallest pixel size at $2x2$ $km^2$ (although this was the "goal" with a "threshold" of $2x3$ $km^2$) while SCIAMACHY had the largest at $30x60km^2$. Regardless of exact numbers, these pixels sizes are orders of magnitude larger than those of GHGSat, but Jacob et al. (2016, Table 2) showed that the proposed missions CarbonSat and GEO-FTS have point source detection thresholds (0.80 and 0.61 tons/hour, respectively) that are much closer to GHGSat (0.25 t/h) that SCIAMACHY (68 t/h) or

GOSAT (7.1 t/h) due to a greater emphasis on measurement precision. An additional sentence somewhere to clarify the differences in precision would enhance understanding for the reader. Furthermore, it might be useful to make one more distinction, the difference between imaging missions (GHGSat, TROPOMI, SCIAMACHY, GeoCarb . . .) and non-imaging missions (GOSAT, MERLIN). Imaging data have clear advantages for point source work, yet the word 'image' never appears in the manuscript, aside from the references.

P2, L1: Can the authors confirm whether 10x10 km$^2$ is indeed correct, since multiple other documents (for example Germain et al., 2017, McKeever et al., 2017 etc.) say 12x12 km$^2$.

P3, L6: Worden et al. (2013) is missing from the reference list.

P4, L3: Subsection 2.2 should be "Source pixel method".

P7, L5-16 and Figure 2. This is a very useful and important result that helps to differentiate between the different challenges in quantifying CH$_4$ and CO$_2$ sources, and the necessary observations and techniques for source estimations.

P8, L27: Additional clarification on the methods of median filtering and Gaussian filtering would be helpful here.

P9, L1-4: The square root of the area seems like a better measure of 'size' than perimeter.

P10, L15-16: The assumption here is that a snapshot of the emissions is representative of the mean annual emission rate, i.e. the intra-annual variability is insignificant or the observation is near the mean of a predictable intra-annual variability, but it is possible that neither of these may be the case depending on the nature of the methane source.

---

## Referee Comment (RC2) · P. Rayner (Referee) · 3 Aug 2018

This paper investigates the utility of a proposed satellite measurement of methane to constrain this gas's point sources. The proposed measurement is unusual for its resolution and geographic focus. The resolution, in particular, imposes demands on the transport models that relate sources and concentrations. The paper investigates this using an ensemble of transport simulations at 50 metre resolution using the Large Eddy Simulation version of WRF. The paper finds that the classical synthesis inversion approach used for mapping sources won't work under these conditions but that a simpler method, using the integrated mass enhancement in a plume, will work well enough.

[Figure]

The paper is well-written and squarely within scope for the journal.

The paper stands in a tradition of work testing the utility of satellite measurements. Like many of these the first essay is almost schematic. This is quite right; their task is to set the conditions under which the proposed measurements can achieve their objective. This paper goes a bit further by comparing methods for using the measurements. This is its most important contribution beyond establishing that the measurement concept is worth pursuing.

the paper does not yet provide convincing evidence that the proposed measurement will, in fact, meet its objectives. There are many questions still to be answered both about the measurement and its interpretation before we can say that. What is the role of pressure, elevation and scattering fluctuations on the mass estimates given that there is no oxygen measurement to normalise photon paths? What will happen when, inevitably, certain measurements are missing from a plume? What is the role of correlated error in differentiating plume from background and calculating uncertainty in mass enhancement? How sensitive is the IME to uncertainties in windspeed and how confident can we be of the extrapolation from surface to effective windspeed in the many combinations of plume elevation and shear that obtain in the real world? the paper does not need to answer any of these but it should open the questions. I request therefore a significantly expanded discussion/conclusions section in which these questions (and I'm sure there are others) can be at least raised, preferably with some suggestions for how they can be addressed.

---

## Author Comment (AC1) · 11 Sep 2018

We thank the reviewer for his comments and suggestions, which we address below. Page and line numbers in our responses refer to the revised manuscript.

1. . . . to make this work most widely applicable to the scientific community, it would be advisable to keep the paper's title more general (removing 'GHGSat').

Response: Thank you for this suggestion. We have removed 'GHGSat' from the title, which now reads, "Quantifying methane point sources from fine-scale satellite observations of atmospheric methane plumes."

[Figure]

2. P1, L25: With such a small number of nadir methane column observing satellites, which use a diversity of technologies that result in a range of observing characteristics, the word "conventional" does not really apply. "Most existing and upcoming methane observing satellites . . ." would be a better introductory phrase.

Response: We agree that the recommended wording is better and have implemented the change (P1, L27).

3. P1, L26: Since Jacob et al. (2016) reviews methane observations from space, the authors could easily have provided a more accurate description of SWIR mission pixel resolutions here than "1-10 km". From the list in Jacob et al. (2016) the proposed CarbonSat has the smallest pixel size at 2x2 km2 (although this was the "goal" with a "threshold" of 2x3 km2) while SCIAMACHY had the largest at 30x60km2. Regardless of exact numbers, these pixels sizes are orders of magnitude larger than those of GHGSat, but Jacob et al. (2016, Table 2) showed that the proposed missions CarbonSat and GEO-FTS have point source detection thresholds (0.80 and 0.61 tons/hour, respectively) that are much closer to GHGSat (0.25 t/h) that SCIAMACHY (68 t/h) or GOSAT (7.1 t/h) due to a greater emphasis on measurement precision. An additional sentence somewhere to clarify the differences in precision would enhance understanding for the reader.

Response: We have added sentences clarifying the differences in column precision and spatial resolution between GHGSat and previous missions (P6, L9-10; P13, L14-17). For further details on previous and upcoming satellite missions, we refer the reader to Jacob et al. 2016, which we cite heavily throughout the text.

4. Furthermore, it might be useful to make one more distinction, the difference between imaging missions (GHGSat, TROPOMI, SCIAMACHY, GeoCarb . . .) and non-imaging missions (GOSAT, MERLIN). Imaging data have clear advantages for point source work, yet the word 'image' never appears in the manuscript, aside from the references.

Response: Thank you for this suggestion. We have added sentences addressing this distinction (P1, L30; P13, L16).

5. P2, L1: Can the authors confirm whether 10x10 km2 is indeed correct, since multiple other documents (for example Germain et al., 2017, McKeever et al., 2017 etc.) say 12x12 km2.

Response: The GHGSat-D demonstration instrument does indeed target 12x12 km2 scenes, but future instruments in the constellation (e.g., GHGSat-C1, to be launched in 2019) will have slightly different scene sizes, depending on orbit altitude and the instrument specifications, which are subject to change. For clarity, we have included the 12x12 km2 figure for GHGSat-D in the manuscript (P6, L8).

6. P3, L6: Worden et al. (2013) is missing from the reference list.

Response: Thank you for catching this oversight. We have added Worden et al. (2013) to the reference list.

7. P4, L3: Subsection 2.2 should be "Source pixel method".

Response: Agreed; we have corrected this error.

8. P8, L27: Additional clarification on the methods of median filtering and Gaussian filtering would be helpful here.

Response: We have added two sentences clarifying our filtering approach (P9, L2-4).

9. P10, L15-16: The assumption here is that a snapshot of the emissions is representative of the mean annual emission rate, i.e. the intra-annual variability is insignificant or the observation is near the mean of a predictable intra-annual variability, but it is possible that neither of these may be the case depending on the nature of the methane source.

Response: We don't intend to make that assumption and now clarify that the retrieval is for an instantaneous source.

---

## Author Response (AR1)

**Responses to reviewers: "Quantifying methane point sources from fine-scale (GHGSat)** satellite observations of atmospheric methane plumes"**

We thank the reviewers for their comments and suggestions, which we address below. Page and line numbers in our responses refer to the revised manuscript.

**Response to comments from Ray Nassar**

1. ... to make this work most widely applicable to the scientific community, it would be advisable to keep the paper's title more general (removing 'GHGSat').

Thank you for this suggestion. We have removed 'GHGSat' from the title, which now reads, "Quantifying methane point sources from fine-scale satellite observations of atmospheric methane plumes."

2. P1, L25: With such a small number of nadir methane column observing satellites, which use a diversity of technologies that result in a range of observing characteristics, the word "conventional" does not really apply. "Most existing and upcoming methane observing satellites . . ." would be a better introductory phrase.

We agree that the recommended wording is better and have implemented the change (P1, L27).

3. P1, L26: Since Jacob et al. (2016) reviews methane observations from space, the authors could easily have provided a more accurate description of SWIR mission pixel resolutions here than "1-10 km". From the list in Jacob et al. (2016) the proposed CarbonSat has the smallest pixel size at 2x2 km2 (although this was the "goal" with a "threshold" of 2x3 km2) while SCIAMACHY had the largest at 30x60km2. Regard- less of exact numbers, these pixels sizes are orders of magnitude larger than those of GHGSat, but Jacob et al. (2016, Table 2) showed that the proposed missions Carbon- Sat and GEO-FTS have point source detection thresholds (0.80 and 0.61 tons/hour, respectively) that are much closer to GHGSat (0.25 t/h) that SCIAMACHY (68 t/h) or GOSAT (7.1 t/h) due to a greater emphasis on measurement precision. An additional sentence somewhere to clarify the differences in precision would enhance understanding for the reader.

We have added sentences clarifying the differences in column precision and spatial resolution between GHGSat and previous missions (P6, L9-10; P13, L14-17). For further details on previous and upcoming satellite missions, we refer the reader to Jacob et al. 2016, which we cite heavily throughout the text.

4. Furthermore, it might be useful to make one more distinction, the difference between imaging missions (GHGSat, TROPOMI, SCIAMACHY, GeoCarb...) and non-imaging missions (GOSAT, MERLIN). Imaging data have clear advantages for point source work, yet the word 'image' never appears in the manuscript, aside from the references.

Thank you for this suggestion. We have added sentences addressing this distinction (P1, L30; P13, L16).

5. P2, L1: Can the authors confirm whether 10x10 km2 is indeed correct, since multiple other documents (for example Germain et al., 2017, McKeever et al., 2017 etc.) say 12x12 km2.

The GHGSat-D demonstration instrument does indeed target 12x12 km2 scenes, but future instruments in the constellation (e.g., GHGSat-C1, to be launched in 2019) will have slightly different scene sizes, depending on orbit altitude and the instrument specifications, which are subject to change. For clarity, we have included the 12x12 km2 figure for GHGSat-D in the manuscript (P6, L8).

6. P3, L6: Worden et al. (2013) is missing from the reference list.

Thank you for catching this oversight. We have added Worden et al. (2013) to the reference list.

7. P4, L3: Subsection 2.2 should be "Source pixel method".

Agreed; we have corrected this error.

8. P8, L27: Additional clarification on the methods of median filtering and Gaussian filtering would be helpful here.

We have added two sentences clarifying our filtering approach (P9, L2-4).

9. P10, L15-16: The assumption here is that a snapshot of the emissions is representative of the mean annual emission rate, i.e. the intra-annual variability is insignificant or the observation is near the mean of a predictable intra-annual variability, but it is possible that neither of these may be the case depending on the nature of the methane source.

We don't intend to make that assumption and now clarify that the retrieval is for an instantaneous source.

**Response to comments from Peter Rayner**

The paper does not yet provide convincing evidence that the proposed measurement will, in fact, meet its objectives. There are many questions still to be answered both about the measurement and its interpretation before we can say that. What is the role of pressure, elevation and scattering fluctuations on the mass estimates given that there is no oxygen measurement to normalize photon paths? What will happen when, inevitably, certain measurements are missing from a plume? What is the role of correlated error in differentiating plume from background and calculating uncertainty in mass enhancement? How sensitive is the IME to uncertainties in wind speed and how confident can we be of the extrapolation from surface to effective wind speed in the many combinations of plume elevation and shear that obtain in the real world? the paper does

not need to answer any of these but it should open the questions. I request therefore a significantly expanded discussion/conclusions section in which these questions (and I'm sure there are others) can be at least raised, preferably with some suggestions for how they can be addressed.

Thank you for these thoughtful questions. We agree that an expanded discussion and conclusions section touching on these and other questions would be valuable to the reader. We have added discussion of these topics to the conclusions section (P13, L18-27).

**List of relevant changes made in the manuscript**

Beyond the changes outlined above, we have made some other minor ones:

- 1. The claim that point sources  $Q \ge 0.5$  t/h account for 75% of emissions reported to the GHGRP was made in error. We corrected it to  $Q \ge 0.3$  t/h.
- 2. We made small formatting and diction changes throughout the text.
- 3. We reproduced the figures in higher resolution and with sans-serif font.
- 4. We expanded the acknowledgements section.

**Deleted: (GHGSat)**

**Quantifying methane point sources from fine-scale satellite observations of atmospheric methane plumes**

Daniel J. Varon1,2, Daniel J. Jacob1, Jason McKeever2, Dylan Jervis2, Berke O. A. Durak2, Yan Xia3, Yi Huang3

1School of Engineering and Applied Sciences, Harvard University, Cambridge, MA 02138, USA
 2GHGSat, Inc., Montréal, QC H2W 1Y5, Canada
 3Department of Atmospheric and Oceanic Sciences, Montréal, QC H3A 0B9, Canada
 *Correspondence to*: Daniel J. Varon (danielvaron@g.harvard.edu)

Abstract. Anthropogenic methane emissions originate from a large number of relatively small point sources. The planned GHGSat satellite fleet aims to quantify emissions from individual point sources by measuring methane column plumes over selected ~10×10 km2 domains with ≤ 50×50 m2 pixel resolution and 1-5% measurement precision. Here we develop algorithms for retrieving point source rates from such measurements. We simulate a large ensemble of instantaneous methane column plumes at 50×50 m2 pixel resolution for a range of atmospheric conditions using the Weather Research and Forecasting model (WRF) in large eddy simulation (LES) mode and adding instrument noise. We show that standard methods to infer source rates by Gaussian plume inversion or source pixel mass balance are prone to large errors because the turbulence expert the neurophysical and the neuroll order of instantaneous methane. The interact of when experiment the neurophysical and n

- cannot be properly parameterized on the small scale of instantaneous methane plumes. The integrated mass enhancement (IME) method, which relates total plume mass to source rate, and the cross-sectional flux method, which infers source rate from fluxes across plume transects, are better adapted to the problem. We show that the JME method with local measurements of the 10-m wind speed can infer source rates with error of 0.07-0.17 t  $h^{-1}$  + 5-12% depending on instrument precision (1-5%).
- 20 The cross-sectional flux method has slightly larger errors (0.07-0.26 t h-1 + 8-12%) but a simpler physical basis. For comparison, point sources larger than 0.2 t h-1 contribute more than 75% of methane emissions reported to the U.S. Greenhouse Gas Reporting Program. Additional error applies if local wind speed measurements are not available, and may dominate the overall error at low wind speeds. Low winds are beneficial for source detection but detrimental for source quantification.

**1** Introduction**

- 25 Satellite instruments can measure atmospheric methane columns from solar backscatter in the shortwave infrared (SWIR) with near-uniform sensitivity down to the surface (Frankenberg et al., 2005). There is considerable interest in using these measurements to quantify methane emissions (Jacob et al., 2016). Most current and planned instruments have pixel resolutions of 1-10 km and column precisions of 0.1-1% (Bovensmann et al., 1999; Butz et al., 2011; Veefkind et al., 2012; Polonsky et al., 2014; Kuze et al., 2016). Jacob et al. (2016) show that these measurements can successfully map regional methane
- 30 emissions but have limited ability to resolve individual methane point sources, even with imaging capabilities, because the

1

|   | Deleted: IME                                                                                                                                                             |
|---|--------------------------------------------------------------------------------------------------------------------------------------------------------------------------|
|   |                                                                                                                                                                          |
| { | Deleted: 5                                                                                                                                                               |
|   | Deleted: not                                                                                                                                                             |
|   |                                                                                                                                                                          |
|   | Deleted: Conventional methane observing satellites are limited in their ability to detect individual point sources. They observe vertical relevance $e^{i\theta}$ |

| Do | otode | hu  |
|----|-------|-----|
|    |       | 111 |

| -{ | Deleted: are adequate for mapping |
|----|-----------------------------------|
| -{ | Deleted: cannot                   |
| -( | Deleted: which                    |

sources tend to be relatively small and spatially clustered (e.g., oil/gas fields, livestock operations, landfills, coal mine vents). The GHGSat microsatellite fleet (Germain et al., 2017; McKeever et al., 2017) aims to address this gap by observing methane columns over selected scenes of order  $10 \times 10 \text{ km}^2$  with  $\leq 50 \times 50 \text{ m}^2$  effective pixel resolution and moderate precision (1-

5%). Here we present algorithms for interpreting the instantaneous plumes observed by such an instrument in terms of the
implied point source (facility-level) emissions, and estimate the associated errors and detection limits as a function of instrument precision.

Aircraft remote sensing of methane columns over oil/gas and coal mining facilities shows that the instantaneous plumes have irregular shapes and detectable sizes of order 0.1-1 km (Thorpe et al., 2016; Thompson et al., 2015; 2016; Frankenberg et al., 2016). A standard method to retrieve source rates from plume observations is to assume Gaussian plume

- 10 behaviour, as expected from statistically averaged turbulence (Bovensmann et al., 2010; Krings et al., 2011; 2013; Rayner et al., 2014; Fioletov et al., 2015; Nassar et al., 2017; Schwandner et al., 2017). This method may induce large errors for small instantaneous plumes, which generally do not follow the steady-state Gaussian behaviour. Several authors have addressed this difficulty. Krings et al. (2011, 2013) proposed a cross-sectional flux method to derive the source rate as the product of the local wind and the concentration integrated over a plume cross-section, expanding on a similar method used for in situ plume measurements (White et al., 1976; Cambaliza et al., 2014; Conley et al., 2016). Jacob et al. (2016) described a mass balance
- method for inferring the source rate solely based on the enhancement in the source pixel. Frankenberg et al. (2016) inferred the source rate empirically from the total detectable mass of methane in the plume (integrated mass enhancement or IME).

A common feature of all these methods for retrieving the point source rate Q from plume observations is their need for independent knowledge of the wind speed U driving transport of the plume. In the cross-sectional flux method applied to

20 in situ aircraft observations, methane and local wind speed are measured concurrently (Conley et al., 2016). In remote sensing, however, the wind speed for the instantaneous column plume is not directly measured and may be variable both vertically and horizontally across the plume.

Here we use observing system simulation experiments to develop algorithms for retrieving individual point source rates from fine-scale satellite observations of instantaneous methane plumes. We review previously-used plume inversion

25 methods and show with large eddy simulations (LES) that the JME and cross-sectional flux methods are best-suited to the problem. We further develop the JME method to provide a physical basis for its general application. We consider different combinations of instrument precision, meteorological environment, and wind information to test the methods and quantify errors. Our work is motivated by GHGSat but is more generally applicable to any fine-scale plume observations from space.

**2 Review of methods for retrieving point sources from observations of column plumes**

30 A methane point source produces a turbulent plume of atmospheric methane with characteristics determined by the strength of the source, the wind field, and turbulence that depends on atmospheric stability and surface roughness. Four different methods have been proposed to quantify point source rates from plume observations: (1) the Gaussian plume inversion method (Bovensmann et al., 2010; Krings et al., 2011; 2013; Rayner et al., 2014; Fioletov et al., 2015; Nassar et al., 2017; Schwandner

| •  |  |  |
|----|--|--|
| ,  |  |  |
| Ζ. |  |  |
|    |  |  |

| - | Deleted: | an | algorithm |
|---|----------|----|-----------|
|---|----------|----|-----------|

| Deleted: reveals |  |
|------------------|--|
| Deleted: with    |  |
|                  |  |

**Deleted: an algorithm**

| Deleted: IME                   |
|--------------------------------|
| Deleted: IME                   |
| Deleted: error                 |
| Deleted: the need to interpret |
| Deleted: observations          |

| 1 |          |            |
|---|----------|------------|
|   | Deleted: | of methane |

et al., 2017), (2) the source pixel method (Jacob et al., 2016; Buchwitz et al., 2017), (3) the cross-sectional flux method (White et al., 1976; Conley et al., 2016; Krings et al., 2011; 2013; Tratt et al., 2011; 2014; Frankenberg et al., 2016), and (4) the JME method (Thompson et al., 2016; Frankenberg et al., 2016). Here we discuss these methods for remote sensing observations of column plumes, This is a somewhat different problem than for in situ observations of plumes. In situ observations benefit from

5 a stronger signal but require characterization of the plume in the vertical dimension, which is integrated in a column measurement.

**Vhite Deleted: IME Image: Image of the system Image of the system Deleted: Image of the system Image of the syste**

(2)

[revised manuscript text omitted]

number  $Pe = UL/K_H$ , where  $K_H$  is the turbulent horizontal diffusion coefficient (Brasseur and Jacob, 2017). For a typical  $K_H = 50 \text{ m}^2 \text{ s}^{-1}$  (d'Isidoro et al., 2010) with  $U = 2 \text{ m} \text{ s}^{-1}$  and L = 50 m we find  $Pe \sim 1$ , so that turbulent diffusion and advection are of comparable importance.

**5 Computing the source rate by the JME method**

5 We showed in Sect. 2.4 how the JME method for retrieving the point source rate Q from the measured JME hinges on knowledge of the residence time of methane in the detectable plume. We refer to this residence time as the plume lifetime  $\tau = IME/Q$ , which in turn is related to two parameters: an effective wind speed  $\mathcal{J}_{eff}$  and a characteristic plume size  $L.\mathcal{J}ME$  and L can be inferred from the plume observations, while  $\mathcal{J}_{eff}$  can be inferred from the observable 10-m wind speed  $U_{10}$  at the point of emission.

**10 5.1 Inferring the plume mass (IME) and size (L)**

Inferring JME and *L* from the plume observations requires that we define the horizontal extent of the plume through a pixel selection procedure that separates signal from noise. Careful selection is important. Consider an array of *N* pixels of equal area and with retrieved column enhancements  $\Delta \Omega_{j_R}(j_R = 1 \dots N_R)$ . If each pixel enhancement includes a contribution  $s_j$  from signal (actual plume enhancement) and  $\varepsilon_j$  from random noise, then as per Eq. (7),

15
$$\frac{\mathrm{IME}}{A} = \sum_{j=1}^{N} \Delta \Omega_j = \sum_{j=1}^{N} (s_j + \varepsilon_j) = \varepsilon_a + \sum_{j=1}^{N} s_j$$
,

where  $\varepsilon_a$  is the total measurement error. The relative error  $\varepsilon_r$  is then  $\varepsilon_r = \varepsilon_a / \sum_j^N s_j$ . If the noise is normally distributed and uncorrelated, then the error standard deviation is proportional to  $\sqrt{N}$ , so that the standard deviation  $\sigma_r$  of the relative error scales as

$$\sigma_r \propto \frac{\sqrt{N}}{\sum_i^N s_j}.$$
 (10)

20 Now consider two extreme cases: (1) all pixels contain the same signal  $s_0$ , and (2) only one pixel contains signal  $s_0$  and the other pixels contain only noise. In case (1), the total signal  $\sum_{j}^{N} s_j$  is proportional to *N*, meaning  $\sigma_{\varepsilon_r} \propto 1/\sqrt{N}$ . By contrast, in case (2), the total signal is equal to  $s_0$ , so  $\sigma_{\varepsilon_r} \propto \sqrt{N}$ . Thus, we see that aggregating plume pixels can either decrease or increase the error on the JME depending on whether these pixels have significant signal or not.

Figure (3) illustrates how we construct a plume mask to select plume pixels with significant signal-noise ratios. The background distribution (mean ± standard deviation) is first characterized by an upwind sample of the measured columns, mimicking what one would do with actual observations. Next, we sample the 5×5 pixels neighbourhood centred on each pixel in the viewing domain and compare the sample distributions to the background distribution by means of a Student's *t*-test. Pixels whose 5×5 neighbourhoods follow a distribution significantly different than the background at a significance level of

| 0        |  |
|----------|--|
| o |  |
|          |  |

| Dele | ted: 100              |
|------|-----------------------|
| Dele | ted: 2                |
| Dele | ted: 50               |
| Dele | ted: IME              |
| Dele | ted: IME              |
| Dele | ted: IME              |
| Dele | ted: U eff |
| Dele | ted: IME              |
| Dele | ted: U eff |

| Deleted: IME |  |  |
|--------------|--|--|
|              |  |  |
|              |  |  |
|              |  |  |
|              |  |  |

| 1  | Deleted: (            |
|----|-----------------------|
| -1 | Deleted: = 1,, |
| ٦  | Deleted: ).           |

(9)

[revised manuscript text omitted]

Figure (7) shows the resulting relationship between  $U_{eff}$  and  $U_{10}$ . The relationship is near-linear, as would be expected, and the fit  $U_{eff} = \beta U_{10}$  with  $\beta = 1.4-1.5$  (where the range is for the 1-5% range of instrument precisions) captures  $20-\frac{75}{9}$  of the variance ( $0.20 \le R^2 \le 0.75$ ) for  $U_{10} \ge 2$  m s-1, depending on instrument precision. The 40-50% increase relative to  $U_{10}$  reflects the increase of wind speed with altitude where the plume is transported. The departure from the linear

relationship for  $U_{10} < 2 \text{ m s}^{-1}$  is because low winds are more variable in direction. The cross-sectional flux method should not 5 be used under calm wind conditions.

Figure (8) shows the results of the cross-sectional flux retrieval algorithm applied to the LES test plumes, excluding those from the plume population with  $U_{10} < 2 \text{ m s}^{-1}$  and  $U_{eff} < 2 \text{ m s}^{-1}$ . In all instrument precision scenarios, the retrieved source rates are consistent with the 1:1 line. However, residuals are slightly larger than in the JME method (see Figure (6)), as indicated by the smaller coefficients of determination. This results primarily from greater uncertainty in the effective wind speed compared to the JME method. Moreover, analysing orthogonal plume cross-sections requires estimation of the wind direction, which introduces an additional source of error. Absolute and relative retrieval errors estimated in the same way as for the JME method are listed in Table 1. While retrieval uncertainty is slightly higher (0.07-0.26 t h-1 + 8-12%, depending on instrument precision), an advantage of the cross-sectional flux method is that there is a simpler physical basis for relating  $U_{10}$ . C, and Q.

**7 Inferring the effective wind speed from meteorological databases**

| Deleted: U e | eff                |  |
|-------------------------|--------------------|--|
| Deleted: U              | ff                 |  |
| Deleted: 1.             | $44 \pm 0.04$      |  |
| Deleted: 77             | /                  |  |
| Deleted: 0.             | $20 \le R^2$       |  |
| Deleted: 0.             | 77                 |  |
| Deleted: 2              |                    |  |
| Deleted: ~              | 44                 |  |
| Deleted: 2              |                    |  |
| Deleted: 2              |                    |  |
| Deleted: U              | eff < 2 |  |
| Deleted: IN             | ИE                 |  |
| Deleted: IN             | ИE                 |  |
| B . I . I . I           | aluzina            |  |

| ocal wind speed              |
|------------------------------|-----------------------|
| ogical database.             |                       |
| Modelling and                |
| m 2 ) at a lowest |                       |
| The 10-m wind                |                       |
|                              |                       |
|                              |                       |
| (13)                         |                       |
|                              |                       |
| <li>(l) is a stability</li>  |
| FP data include              |
| er databases than            |                       |
| used as a global             |                       |
|                              |                       |
| s in June 2017 at            |                       |
| e we use $z_{0,m} =$         |
|                              |                       |
|                              |                       |

Both the JME and cross-sectional flux methods require knowledge of the local wind speed. In the absence of measurements, the 10-m wind speed  $U_{10}$  at the time of observation must be estimated from some meteorol Here we examine the option of using the GEOS-FP operational reanalysis produced by the NASA Global

Assimilation Office, available globally as 3-hour averages with 0.25°×0.3125° resolution (≈ 25×25 km gridpoint level of 60 m above the surface (Molod et al., 2012; https://gmao.gsfc.nasa.gov/GMAO\_products/). speed can be obtained from the 60-m wind speed by:

$$U_{10} = \left[ \frac{\ln\left(\frac{z_{10}}{z_{0,m}}\right) - \Psi_m}{\ln\left(\frac{z_{60}}{z_{0,m}}\right) - \Psi_m} \right] U_{60} , \qquad (13)$$

10

15

20

where  $z_{0,m}$  [m] is the surface roughness length for momentum,  $z_{10} = 10$  m,  $z_{60} = 60$  m, and  $\Psi_m = f(z_m)$ 25 correction parameter dependent on the Monin-Obukhov length l (Brasseur and Jacob, 2017). The GEOSvalues for  $z_{0,m}$  and l, but one can use local estimates of these variables if better information is available. Better GEOS-FP may be available to the user depending on region, but an advantage of GEOS-FP is that it can be default.

Figure (9) evaluates the GEOS-FP  $U_{10}$  data by comparison to 3-hour average daytime measurements 30 10 U.S. airports obtained from the University of Utah MesoWest database (http://mesowest.utah.edu/). Her 0.025 m as input to Eq. (13) to account for the relatively smooth airport terrain. There is no bias in the GEOS-FP data relative to MesoWest. The error standard deviation derived from the difference between the 3-hour GEOS-FP and MesoWest 10-m wind speeds is  $\frac{1.6}{1.6}$  m s-1, largely independent of wind speed. Since wind speed is a positive variable, errors at low wind speeds (<2 m s-1) tend to be systematic. There is additional error from using 3-hour wind data when the plume lifetime  $\tau$  is much

5 shorter. From the 5-minute resolution of the MesoWest data we find an additional error standard deviation of 2.0 m s-1 for  $\tau = 1$  for  $\tau = 1$  hour when 3-h average wind speed data are used. Adding these errors in quadrature, we conclude that using GEOS-FP wind data incurs an error standard deviation on the 10-m wind speed of 2.5 m s-1 for small plumes ( $\tau = 5$  minutes) and 2.0 m s-1 for large plumes ( $\tau = 1$  hour).

Substitution into the  $U_{\text{eff}} = f(U_{10})$  relations of the JME and cross-sectional flux methods implies an additional error in inferring Q of 15-50% for the JME method and 30-65% for the cross-sectional flux method over the 10-m wind speed range 2-7 m s-1, with largest errors at low wind speeds. The error is larger for the cross-sectional flux method where the dependence of  $U_{\text{eff}}$  on  $U_{10}$  is linear rather than logarithmic. Comparison to the other retrieval errors for each method is given in Table 1. At low wind speeds, the error from using GEOS-FP wind data may dominate the overall error budget for inferring source rates. However, our estimate of the error from using operational meteorological databases is intended only to be illustrative. Different

15 errors may apply for other regions or seasons, or when using other meteorological databases than GEOS-FP.

**8 Conclusions**

We have developed new algorithms for quantifying methane point sources from fine-scale satellite observations of atmospheric column plumes, motivated by the planned fleet of GHGSat instruments ( $\leq 50 \times 50 \text{ m}^2$  pixel resolution, 1-5% precision). A challenge is that individual point sources of methane are relatively weak, so that the detectable instantaneous plumes are

- 20 relatively small (~1 km) and short-lived (<1 hour). Using a large ensemble of WRF large eddy simulations (LES) of methane plumes from point sources, we showed that Gaussian plume inversions are unsuccessful because the instantaneous plumes are too small to follow Gaussian behaviour. We also showed how a simple source pixel mass balance method is inappropriate because of wind variability and horizontal turbulent diffusion on the scales of relevance.
- Two more promising methods for quantifying source rates from methane column plume observations are the 25 integrated mass enhancement ( $\underline{IME}$ ) method and the cross-sectional flux method. Both methods require construction of a plume mask to isolate the plume enhancements from the background noise. The JME method requires estimation of the plume lifetime  $\tau$ , which in turn depends on an effective wind speed  $\underline{J}_{eff}$  for the plume and a characteristic plume size *L*. We showed how these quantities can be estimated from knowledge of the plume mask and of the 10-m wind speed  $U_{10}$  at the location of the source. The source rates are then inferred from the plume observations with expected errors of 0.07-0.17 t h-1 + 5-12%
- 30 depending on instrument precision (1-5%). For reference, source rates larger than  $0\frac{3}{4}$  t h-1 contribute more than 75% of total point source emissions in the U.S. Greenhouse Gas Reporting Program (GHGRP) database.

[revised manuscript text omitted]